# Breast Cancer Cell Type and Biomechanical Properties of Decellularized Mouse Organs Drives Tumor Cell Colonization

**DOI:** 10.3390/cells12162030

**Published:** 2023-08-09

**Authors:** Anton D. Pospelov, Olga M. Kutova, Yuri M. Efremov, Albina A. Nekrasova, Daria B. Trushina, Sofia D. Gefter, Elena I. Cherkasova, Lidia B. Timofeeva, Peter S. Timashev, Andrei V. Zvyagin, Irina V. Balalaeva

**Affiliations:** 1Institute of Biology and Biomedicine, Lobachevsky State University of Nizhny Novgorod, 23 Gagarin Ave., Nizhny Novgorod 603950, Russia; eso103163@gmail.com (A.D.P.); kutovaom@gmail.com (O.M.K.); sofia.gieftier.00@mail.ru (S.D.G.); cherkasova.el@yandex.ru (E.I.C.); bioli@mail.ru (L.B.T.); andrei.zvyagin@mq.edu.au (A.V.Z.); 2Shemyakin-Ovchinnikov Institute of Bioorganic Chemistry of the Russian Academy of Sciences, Miklukho-Maklaya, 16/10, Moscow 117997, Russia; timashev_p_s@staff.sechenov.ru; 3Institute for Regenerative Medicine, Sechenov University, Moscow 117418, Russia; efremov_yu_m@staff.sechenov.ru (Y.M.E.); albina.nekrasova2012@yandex.ru (A.A.N.); 4Phystech School of Biological and Medical Physics, Moscow Institute of Physics and Technology, Dolgoprudny 141701, Russia; 5Federal Research Center Crystallography and Photonics, Russian Academy of Sciences, Moscow 119991, Russia; trushina.d@mail.ru; 6Institute of Molecular Theranostics, Sechenov First Moscow State Medical University, Moscow 119435, Russia; 7Privolzhsky Research Medical University, 10/1, Minin and Pozharsky Sq., Nizhny Novgorod 603950, Russia; 8Chemistry Department, Lomonosov Moscow State University, Leninskiye Gory 1–3, Moscow 119991, Russia; 9Laboratory of Clinical Smart Nanotechnology, Sechenov University, Moscow 117418, Russia

**Keywords:** tumor microenvironment, extracellular matrix, metastatic process, decellularization, 3D tumor models, breast cancer, matrix stiffness

## Abstract

Tissue engineering has emerged as an indispensable tool for the reconstruction of organ-specific environments. Organ-derived extracellular matrices (ECM) and, especially, decellularized tissues (DCL) are recognized as the most successful biomaterials in regenerative medicine, as DCL preserves the most essential organ-specific ECM properties such as composition alongside biomechanics characterized by stiffness and porosity. Expansion of the DCL technology to cancer biology research, drug development, and nanomedicine is pending refinement of the existing DCL protocols whose reproducibility remains sub-optimal varying from organ to organ. We introduce a facile decellularization protocol universally applicable to murine organs, including liver, lungs, spleen, kidneys, and ovaries, with demonstrated robustness, reproducibility, high purification from cell debris, and architecture preservation, as confirmed by the histological and SEM analysis. The biomechanical properties of as-produced DCL organs expressed in terms of the local and total stiffness were measured using our facile methodology and were found well preserved in comparison with the intact organs. To demonstrate the utility of the developed DCL model to cancer research, we engineered three-dimensional tissue constructs by recellularization representative decellularized organs and collagenous hydrogel with human breast cancer cells of pronounced mesenchymal (MDA-MB-231) or epithelial (SKBR-3) phenotypes. The biomechanical properties of the DCL organs were found pivotal to determining the cancer cell fate and progression. Our histological and scanning electron microscopy (SEM) study revealed that the larger the ECM mean pore size and the smaller the total stiffness (as in lung and ovary), the more proliferative and invasive the mesenchymal cells became. At the same time, the low local stiffness ECMs (ranged 2.8–3.6 kPa) did support the epithelial-like SKBR-3 cells’ viability (as in lung and spleen), while stiff ECMs did not. The total and local stiffness of the collagenous hydrogel was measured too low to sustain the proliferative potential of both cell lines. The observed cell proliferation patterns were easily interpretable in terms of the ECM biomechanical properties, such as binding sites, embedment facilities, and migration space. As such, our three-dimensional tissue engineering model is scalable and adaptable for pharmacological testing and cancer biology research of metastatic and primary tumors, including early metastatic colonization in native organ-specific ECM.

## 1. Introduction

The idea of the tumor microenvironment as a pivotal factor that determines the behavior of cancer cells has become the central point in modern cancer biology. The tumor microenvironment includes cells of different origin and functions: stromal cells, immune cells, endotheliocytes, etc., and non-cellular component—the extracellular matrix (ECM). ECM is intricately involved in the maintenance of tissue-specific homeostasis [1,2]. The chemical composition of ECM, that is, the type and the ratio of proteins, polysaccharides, and water, and its physical properties arising from the organization of these components, are optimal to each tissue or organ. ECM defines tissue architecture serving as the scaffold and the dynamic regulator of vital processes of the cells [3,4]. Changes in the composition and properties of the ECM are directly involved in tissue transformation and carcinogenesis [5,6]. The role of ECM in tumor development is being actively studied from both biochemical [7,8] and biomechanical perspectives [9,10,11]. On artificial 2D substrates, it has been widely reported that cancer cells prefer stiffer substrates and, moreover, cause stiffening of the surrounding ECM themselves by means of its chemical modification which in turn reinforce cell growth and invasion [9,10,12]. However, the physicochemical properties of artificial substrates significantly differ from that of the native tissue matrices and lack the characteristic 3D architectonics which crucially influences the cells’ behavior [13,14].

Fortunately, the field of regenerative medicine has conceptualized a method that allows to obtain a purified extracellular matrix of a given tissue that retains all its native traits by removing the cellular component, that is, tissue decellularization. An appropriate protocol makes it possible to achieve the preservation of both chemical composition and three-dimensional structure, as well as a number of other biomechanical characteristics [15]. Due to the native density and architectonics of matrix fibers, an adequate distribution of the cellular component is anticipated during recellularization. This allows using the obtained material as a highly biocompatible scaffold for creating models of tumor growth [16,17,18], which is especially relevant for studying the cell–cell and cell–matrix interactions [19,20].

In this work, we question if the stiffness preferences of the cancer cells in the 3D DCL model are similar to convenient 2D models, and which biomechanical properties of the native tissue matrix determine the success of its repopulation by cancer cells. The model can shed light on the mechanisms underlying tumor growth, invasion, and metastasis, in particular, the possible role of ECM in forming the pre-metastatic niches. We have created a facile decellularization protocol universally applicable to a spectrum of murine organs which provide a high degree of purification from cell debris and preservation of native architectonics. The obtained DCL matrices show organ-specific structural and mechanical properties, namely, the pore size and organization of fibers as well as stiffness measured at macrolevel (total stiffness) and nanolevel (local stiffness). We have also shown, for the first time, that in natural DCL matrices of normal organs, increasing stiffness hinders their population of breast cancer cells. Of importance, cells of different phenotypes are susceptible to different parameters: mesenchymal breast cancer cells MDA-MB-231 prioritize matrices with high porosity and low total stiffness, while epithelial-like SKBR-3 cells favorize an environment with low local stiffness of the fibers.

## 2. Materials and Methods

### 2.1. Laboratory Animals and Organ Harvesting

Thirty BALB/C mice (females, 4 months old) were obtained from the SPF vivarium of the Lobachevsky State University of Nizhny Novgorod (UNN). The animals were kept under standard conditions (at 25–26 °C and 12 h light–dark cycle) supplied with food and water *ad libitum*. The experiment was approved by the Bioethical Commission of the UNN (protocol No. 15 dated 15 February 2018). Animals were euthanized by cervical dislocation.

The general scheme of the experiment is shown in Figure 1. The euthanized animals were treated with 96% ethanol. The autopsy was performed in a laminar flow cabinet using autoclaved instruments to ensure the asepticity of the procedure and the obtained samples. The liver, kidneys, ovaries, spleen, and lungs were collected from each animal.

### 2.2. Decellularization of the Organ Samples

The original protocol was developed for obtaining cell-free extracellular matrices of the mouse organs. The collected organs were washed in sterile distilled water. Fat adhesions (if any) were removed from each organ, after which the organs were dissected into 0.3 × 0.3 cm pieces (2–5 samples from one organ). Then, the samples were placed in 50 mL tubes with 40 mL of 0.5% solution of Triton X-100 (Sigma-Aldrich, Burlington, MA, USA) for 1 h. Following this, the samples were transferred to the tubes with 40 mL of 0.5% sodium dodecyl sulfate (SDS, Sigma-Aldrich, Burlington, MA, USA) solution, and 40 mL of 1% sodium deoxycholate (SDC, Sigma-Aldrich, Burlington, MA, USA) solution subsequently for 1 h each. After that, the samples were placed in 40 mL of 0.075% SDS solution for 24 h. All decellularization media contained 50 μg/mL penicillin and 50 μg/mL streptomycin (PanEco, Moscow, Russia). At each stage of the protocol, the tubes with samples were fixed on an orbital shaker with a rotation speed of 80 rpm.

### 2.3. Total DNA Content Analysis

To determine the DNA content, the DCL matrices were minced with further incubation in a solution of 0.12% collagenase type II (PanEco, Russia) and 0.08% trypsin in 1 mL of Dulbecco’s Modified Eagle’s medium (DMEM, PanEco, Russia) for 1.5 h at 37 °C. The obtained mixture was centrifuged (400 g, for 10 min) and the pellet was rinsed with 200 µL of phosphate-buffered saline (PBS). Total DNA was isolated using the ExtractDNA Blood DNA purification kit (VM011, Evrogen, Moscow, Russia) according to the manufacturer’s instructions. The concentration of DNA was measured on a NanoVue (GE Healthcare, Chicago, IL, USA) drop spectrophotometer.

### 2.4. Cell Culture

Human breast cancer cell lines MDA-MB-231 (Sigma Aldrich, St. Louis, MO, USA) and SKBR3 (Russian collection of cell cultures, Moscow, Russia) were cultured on DMEM, containing 2 mM glutamine (PanEco, Russia), 10% (*v*/*v*) newborn calf serum (NBCS, Gibco, Te Aroha, New Zealand), 50 μg/mL penicillin, and 50 μg/mL streptomycin (PanEco, Moscow, Russia) at 37 ℃ in 5% CO_2_. For passaging, cells were detached with Versene solution (PanEco, Moscow, Russia).

### 2.5. Recellularization of the Matrices

To wash off the matrices from residual detergents, as well as to saturate the core of the matrix with nutrients, the samples were sequentially incubated in PBS for 4 days with PBS change every 24 h, then in DMEM media for 72 h, and then incubated in DMEM with 30% NBCS for 7 days with a change of medium every 72 h.

Each preconditioned matrix was placed in an individual well of a 6-well ultra-low attachment plate (Corning, Corning, NY, USA). Each matrix was repopulated by injecting 300,000 cells in 500 μL of complete growth media. The cell suspension was distributed inside the matrix by several injections via an insulin syringe with a 0.45 mm needle diameter. After injection, 5 mL of DMEM with 15% serum was added to each well of the plate. Then the matrices were incubated at 37 °C in an atmosphere of 5% CO_2_ for 7 days. On days 3 and 5, the medium bathing of the matrix was collected and centrifuged at 1000× *g* for 5 min. The resulting cell pellet was diluted in a complete growth medium and the suspension was re-injected into matrices. During the preliminary stage of the experiments, we have tried multiple methods of recellularization (dripping, injection, bathing in a cell suspension) and found the injection to be the most applicable for obtained samples, which agrees with other research [21,22].

### 2.6. 3D Tumor Growth Models in Collagen Hydrogel

Collagen hydrogel was prepared as previously described [23]. Hydrogels were obtained in a 6-well plate (Corning, Corning, NY, USA) by thorough mixing: 1600 µL of cooled collagen pre-gel (Type I collagen from rat tails, 1.2 mg/mL, dissolved in 0.1% acetic acid); 450 µL of nutrient-buffer solution containing 10× DMEM (Biowest, Nuaillé, France), 25 mM glutamine, 1 M HEPES (PanEco, Moscow, Russia), and 50% newborn calf serum (NBCS); 3 × 10^5^ cells in 200 µL of DMEM, and 134 µL of 0.34 M NaOH. The gels were incubated at 37 °C in 5% CO_2_ for 15–20 min until complete gelation. Cell concentration was about 1.25 × 10^5^ cells per 1 mL of the gel. After solidification of the hydrogel, 2 mL of full DMEM with 10% NBCS was added to the wells and the hydrogels were detached with the pipette tip from the walls of the plate wells. Hydrogels with embedded cells were incubated at 37 °C in 5% CO_2_; the growth medium was exchanged to a fresh medium every 72 h for 7 days.

For rheological analysis, the collagen hydrogel was prepared in a 2 mL tube by successively mixing the following solutions in analogous proportions: 800 µL of the collagen pre-gel, 225 µL of nutrient-buffer solution, 100 µL of DMEM, and 67 µL of 0.34 M NaOH. The tube with the mixture was placed on the measuring stage of the rheometer right after the addition of NaOH and the measurements were started immediately.

### 2.7. Histomorphological Analysis

The matrices (both decellularized and collagen ones) were removed from the wells of the plate and placed in 10% neutralized formalin (Biovitrum, Saint Petersburg, Russia) for 24 h at room temperature. Then they were washed from excess fixative under running water for 20 min and dehydrated in 5 replicates of modified isopropyl alcohol “Blik” (BlikMediklProduction, Taganrog, Russia) according to the manufacturer’s instructions. The matrices were embedded in paraffin and the sections (2 sections per 1 sample) with a thickness of 7 μm were prepared using Epredia HM 325 microtome (Thermo Fisher Scientific, Waltham, MA, USA). The total amount of analyzed sections for each organ was 80–100. The staining of the slides with hematoxylin–eosin (Mayer’s hematoxylin, Eosin 1% aqueous solution; Biovitrum, Russia) was carried out according to the manufacturer’s instructions. Then, the slides were mounted via Shandon Mounting Media (Thermo Fisher, Waltham, MA, USA) and analyzed on the Zeiss Vert.A1 microscope (Zeiss, Oberkochen, Baden-Württemberg, Germany) with the Zeiss LD A-Plan objectives (Zeiss, Oberkochen, Baden-Württemberg, Germany).

The pore area distribution was analyzed using the ImageJ software (version 1.50i, National Institute of Health, Bethesda, MD, USA). The initial histological image of the matrix was converted to 8-bit image, inverted, and changed to black and white. The area of the individual pores was calculated automatically with the threshold applied to ensure the visual integrity of the pores and low background noise. For more details see Appendix A.

The degree of the repopulation of matrices was estimated as follows. The images of the sections of matrices were obtained (N_sextions_ = 10 for each organ). The number of the cells was counted using the Labscope (version 3.4, Zeiss, Oberkochen, Baden-Württemberg, Germany) software and Annotate & Measure-Count tool in the area of 1 mm^2^ which matched the full image. For qualitative assessment, the following scale was applied: less than 10 cells per 1 mm^2^, “single cells”; 10 to 50 cells per 1 mm^2^, “low level of repopulation“; from 50 to 100 cells per 1 mm^2^, “medium level of repopulation”; more than 100 cells per 1 mm^2^, “high level of repopulation”.

Nuclear–cytoplasmic ratio (N/C) was measured using the ZenLight program (verion 3.0.79.0000, Zeiss, Oberkochen, Baden-Württemberg, Germany). For 30 cells on each field of view (*n* = 5) on each sample (*n* = 10), the area of cytoplasm was measured, then the procedure was repeated with the nuclei. 

### 2.8. Scanning Electron Microscopy

Tissue samples were lyophilized using Labconco freeze drying at −70 °C for 24 h. To eliminate the accumulation of surface charge during scanning with an electron beam in a microscope column, a thin conductive layer of chromium was applied to the samples by physical deposition from the gas phase (PVD-coated Covap, Angstrom of mechanical engineering with an INFICON SKK-310 deposition controller).

The microstructure of decellularized tissues and their morphology were studied using a Hitachi TM4000Plus desktop scanning electron microscope (Hitachi, Marunouchi, Chiyoda-ku, Tokyo, Japan) at a voltage of 10–15 kV in the mode of backscattered electrons. 

The sizes of pores and fibers were measured manually using ImageJ software. On each image, 6 fields were analyzed with 30 pores measured in each field; 5 independently obtained samples of each type of matrix were examined.

### 2.9. Macroindentation

Macroindentation was performed with a Mach-1™ v500csst universal micromechanical system (Biomomentum Inc., Laval, QC, Canada). For the measurements on the matrices, a ruby spherical indenter with a radius of 1 mm was used. Before measurements, a piece of sandpaper was glued to the bottom platform of the device, which further prevented the lateral displacement of the sample during indentation. During the entire measurement, the sample was kept moist by adding PBS. The collagen gels were measured using a metal spherical indenter with a radius of 3.175 mm in the DMEM medium.

Using the instrument’s “Find Contact” function, the exact position of the substrate (sandpaper) was found. This function ensures that the indenter moves until the specified load is registered on the sensor (0.1 N). Subsequently, the sample thickness was calculated relative to the vertical position of the substrate.

Using the same function (“Find Contact“), the sample was indented to a given load (0.1–0.2 N), which ensures an indentation depth of the order of a third to a half of the sample thickness. The obtained dependences of the load (F) on the indentation depth (δ) were approximated by the Hertz model with a correction for the finite sample thickness:$$F=\frac{4}{3}f(\delta )\frac{E}{1-v^{2}}\delta^{\frac{3}{2}}\sqrt{R}$$
where E is Young’s modulus, ν is the Poisson’s ratio of the sample (was taken equal to 0.5 as for most biological samples), and f(δ) is the thickness correction function. Details of applying the model are described in the work [24]. Decellularized matrices (samples) from 10 different animals were analyzed for each type of organ; 3 to 5 measurements were performed in different areas of each sample. 

### 2.10. Atomic Force Microscopy (Nanoindentation)

Nanoindentation experiments were performed using a Bioscope Resolve atomic force microscope (Bruker, Billerica, MA, USA) on sample sections. Frozen sections were obtained from a frozen sample block, embedded in an optimal cutting temperature compound (OCT), using a ThermoScientific HM525NX rotary cryotome (Thermo Fisher, Waltham, MA, USA). The thickness of frozen sections was 20 µm. The sections were transferred on histological slides, then washed from the OCT with PBS. The measurements were conducted in PBS in the Force Volume mode using SAA-HPI custom-modified cantilevers (Bruker, Billerica, MA, USA), which had a spring constant calibrated by the manufacturer (0.186 N/m) and a tip curvature radius of 3.5 μm (hemisphere). In the Force Volume mode, an array of force curves (maps) was taken along a grid ranging in size from 10 × 10 to 100 × 100 µm, consisting of 20 × 20 to 64 × 64 points. In each section, 5–6 such maps were acquired and 5 samples of each type of matrix were examined.

Individual force curves were acquired at a speed of 20 μm/s, the indentation depth was less than 1 μm, thus no correction for sample thickness was required (depth < 5% thickness). The Hertz model was applied for processing, the details of the method and processing are described in the work [24]. The values were then averaged between all force curves of the map and between all maps to get the mean Young’s modulus value and SD of the sample. 

### 2.11. Rheometry

Rheological experiments on collagen hydrogels were performed using a rheometer Physica MCR 302 (Anton Paar, Graz, Austria) in oscillatory mode at room temperature in a high-humidity camera as described in [25]. A plate–plate geometry (25 mm diameter of the upper plate) was used with a gap of 1 mm, which corresponded to 0.5 mL of the sample solution. The collagen pre-gel was placed on the measuring stage right after mixing with 0.34 M NaOH and the measurements were started immediately. Isothermal measurements were then performed with the controlled strain amplitude of 0.5% and frequency of 1 Hz at 25 °C. Tests were repeated three times; storage modulus, G′ (Pa), and loss modulus, G″ (Pa), were measured.

## 3. Results

### 3.1. Obtaining DCL Matrices of Murine Organs

As tissue donors for the matrices, we have selected mice of the BALB/c line. This choice is because murine models are very common in experimental oncology, which makes the use of their matrices more relevant when comparing the matrix-based in vitro models with tumors on animals. There are a few reported protocols tested for mouse tissues. However, almost all of the available protocols were adapted for the decellularization of only 1–2 organs [26,27], which reduces the final output of matrices from one animal and could not be applied to organs with a different structure.

We have previously tested [16] the possibility of decellularization of various murine organs using two protocols of chemical–enzymatic decellularization. Despite the high level of cellular elimination, the integrity of matrices of organs with loose parenchyma was compromised. To overcome the problem of balancing structural integrity and cell-free purity of the resulting matrix, we have proposed here an alternative method of decellularization based on non-ionic detergent and emulsifier Triton X-100, anionic detergents sodium dodecyl sulfate (SDS), and sodium deoxycholate (SDC). The main criteria for the success of the protocol were considered to be a high level of elimination of the cellular component and the preservation of most of the primary histological structures of the organ. The first step of the protocol is an hour of incubation in 0.5% Triton X-100, which allows to quickly destroy the cell membranes of the surface layer and prevents possible activation of necrosis, and, accordingly, the release of hydrolyzing enzymes. The next step we used is incubation in 0.5% SDS, which on the one hand allows us to neutralize the remnants of charged molecules, reducing the adhesion of cells to the matrix, and on the other hand to continue the decomposition of membranes. In the next stage, we used incubation in 1% SDC for an hour. This detergent has a high ability to solubilize membranes and extract membrane proteins without affecting matrix proteins [28]. The final stage of decellularization is 24 h incubation in 0.075% SDS, which finalizes the membrane decomposition. One of the drawbacks when using SDS in high concentrations is the destruction of collagen fibers [29]. In our proposed protocol, an exposition to a high concentration (0.5%) of this detergent is short term, which allows the membranes to be quickly destroyed, followed by a long exposure at a mild-acting low SDS concentration (0.075%). Although this approach is relatively not time consuming (27 h), it allows eliminating the remnants of lipid-associated cellular debris without damaging protein structures. The performance of the proposed protocol for obtaining cell-free matrices with the usage of low detergent concentrations is confirmed by the results of determining the content of residual DNA. In all the studied matrix samples, total DNA concentration decreased by more than 95% (Table 1), which indicates successful decellularization [15,30].

### 3.2. Histological Analysis of Decellularized Matrices

The choice of organs for decellularization (lungs, liver, kidney, spleen, and ovaries) was due to the profile of metastatic activity of breast cancer. According to statistics, the most common metastasis targets of this type of cancer are the lungs, spleen, and liver, and the rarest target is the ovary and kidney [31]. To understand how the proposed method preserves the structural integrity of the matrix, we analyzed histological sections of the initial native tissues and the decellularized matrices obtained from them. It is worth noting that matrices from both non-capsuled (Figure 2A) and capsuled (Figure 2B) organs demonstrated a high quality of preserving architectonics and cell-free purity. We should mention, however, that the proposed protocol is adapted for specimens less than 50 mm^3^, otherwise, residual cells may be found in the core of the matrix. 

#### 3.2.1. Liver

Decellularization of the liver has shown a good quality of decellularization (Figure 2A). The specimen demonstrated a cell-free matrix with good visualization of the pores at the cell sites. At the same time, we could visualize the main structures, such as the portal zone, lobules, and different ducts.

#### 3.2.2. Lung

In the decellularized lung, mild swelling of collagen fibers and the absence of a cellular component was observed (Figure 2A). The typical lung structures, such as alveoli, bronchioles, and supply vessels were recognizable, and their structure was well preserved.

#### 3.2.3. Spleen

After decellularization, the spleen matrix was represented by a network of fibers, in which dense areas in the place of the white pulp and loose areas in the place of the red pulp were distinguishable (Figure 2A). Trabeculae with central arteries were visible in the white pulp. Cellular components were not found.

#### 3.2.4. Kidney

Decellularized kidney matrix provided a full variety of cell-free native structures, such as renal corpuscles, nephron channels, and large blood vessels (Figure 2B). The capsule of the DCL organ was dense and undamaged. Optically empty rounded structures were visible in place of the cells.

#### 3.2.5. Ovary

The resulting ovary matrix was a heterogeneous network of fibers with an uneven fiber density (Figure 2B). After decellularization, voids were observed in place of a part of the follicles and eliminated cells of the follicular epithelium in the place of others. No cells were found in the specimen.

Collagen-based hydrogel is widely used in 3D in vitro models of cancer growth since collagen is the main structural protein in ECM. To evaluate the structural similarity of this kind of artificial scaffold to natural matrices, we analyzed it in the same way. The collagen matrix was represented by a network of weakly basophilic fibers that did not have a clear ordering (Figure 2C and Appendix A). In some places, it was possible to note the compaction of fibers.

Quantitative analysis of the pore area of the matrices of different organs showed that in the lungs, liver, and kidneys, pores with an area of 200–400 µm^2^ predominate, while the spleen and ovaries are characterized by pores with an area below 200 µm^2^ (Figure 3A). The largest average pore area was found for the lungs (20.8 mkm^2^), and the ovary has the smallest pores (4.5 µm^2^). The liver, kidneys, and spleen are characterized by an average pore size of 16.4 µm^2^, 19.8 µm^2^, and 9.11 µm^2^, respectively.

Thus, the histological analysis supported the effective removal of the cellular component and evidenced the preservation of the native three-dimensional organization of the matrix structure for all the organs analyzed in the study.

### 3.3. SEM Analysis of Matrices

To further compare the matrices’ organization, we employed scanning electron microscopy. The analysis of the microstructure has shown a high degree of purification of all the matrices from the cellular component. At the same time, it was determined that the matrices differ in the degree of porosity, fiber thickness, pore shape (e.g., in the matrix of the spleen, the pores are elongated and ordered in parallel, while in the lungs, the pores are rounded hexagonal), and in several other aspects, which makes each organ unique in terms of the possibilities of cell–matrix interaction. 

The liver matrix (Figure 2A) was a homogeneous structure with high porosity of a three-dimensional fibrous scaffold with an average fiber diameter of ~1.6 µm and a pore area of ~133 µm^2^ (Figure 3B,C). The pore area in the lung matrix was significantly larger (~240 µm^2^), with a fiber diameter of ~1.13 µm (Figure 3B,C). The matrix of the spleen, on the contrary, has a denser structure and has the greatest compactification of fibers, and the pore area was ~78 µm^2^ (Figure 2A and Figure 3B,C).

The kidney and ovary are surrounded by a connective tissue capsule, which led to the need to analyze these structures separately. The capsule and parenchyma of the kidney matrix significantly differed in microstructure (Figure 2B). The fibrous capsule of the kidney was a dense structure with evenly distributed holes, the thickness of the capsule could be estimated as ~10 µm. At the same time, the kidney parenchyma was a highly porous network of collagen fibers with a thickness of ~1.31 µm and a pore area of ~90 µm^2^ (Figure 3B,C). The surface of the connective tissue capsule of the ovary was embossed (Figure 2B), and at a higher magnification, the matrix of the surface epithelium with a thickness of fibrils of ~0.73 µm and a pore area of ~1.5 µm^2^ was visible (Figure 3B,C). It should be noted that the properties of kidney and ovarian capsules were different. So, the ovarian capsule was highly fibrillar and dense with a large number of small pores, while in the kidney capsule, the pores were smaller, and their diameter was larger. An important fact is that despite the use of samples obtained by different methods (histological and SEM images), as well as various approaches to analyzing the pore area, the results are comparable, being both within the same orders of values.

### 3.4. Mechanical Properties of the Decellularized Matrices

#### 3.4.1. Macroindentation

The macroindentation measurement allows us to evaluate the macroscale properties of matrices, taking into account the cumulative properties of the organ, including the presence of a dense capsule, matrix porosity, fiber density, chemical composition, and arrangement of fibers in a 3D network. Previously, this method was developed and used to determine the stiffness of bone tissue, however, currently it is used to measure both stiff and soft tissues. It is worth noting, that to the best of our knowledge, this method was rarely used to measure the stiffness of decellularized matrices [32,33,34,35,36]. A decrease in Young’s modulus was observed in the series spleen > kidney > liver > ovary > lung matrices with measured values of ~40 KPa, ~22 KPa, ~20 KPa, ~19 KPa, and ~17 KPa respectively (Figure 4A). This result can be explained by different structural features, such as a fibrillar capsule for ovary and kidney, or parallel fiber package for spleen (Figure 2A,B). Due to a lack of information about matrix stiffness measured with this method, we compared the obtained results with those obtained for human tissues. It was found that the measured matrix stiffness values are similar to the stiffness of intact human organs which was reported previously (spleen ~20 KPa, kidney ~10 KPa, lungs ~12 KPa, liver ~9 KPa) [37]. Overall, the data are consistent with the results of SEM analysis, where a dense fibrillar capsule was observed in the ovary and kidneys. The data obtained on the ovary had the largest variability, which can be connected to the uneven fiber density observed with histology. The high Young’s modulus for the spleen is presumably associated with high fiber compaction. It should be noted that despite the apparent relationship between total stiffness and pore size, no significant negative correlation was found between these parameters (Appendix A).

The same macroindentation technique was used to estimate the stiffness of the artificial collagen hydrogel. The values of Young’s modulus for collagen matrices were substantially lower (Young’s modulus ~0.36 KPa) (Figure 4A). Since the values obtained for hydrogel were close to the border of Biomomentum detection range, we also performed the measurements of the mechanical properties of the collagen matrices by shear rheology. Very similar values (360 ± 190 Pa) were obtained (Appendix A). Thus, in comparison with decellularized matrices, collagen hydrogel, prepared following the protocol commonly used for cellular studies [23], has a rigidity that is at least an order of magnitude lower. Also, an important aspect is the general disorder of the fibers (Figure 2C), which is strikingly different from the native tissue matrices. Summing up, even though collagen hydrogel has chemical characteristics similar to most of the decellularized matrices, collagen-based gels do not have the same structural and biomechanical features, which makes this model less relevant for studies aimed at researching cell–cell and cell–matrix interactions.

#### 3.4.2. Nanoindentation

The second approach to measure stiffness, nanoindentation by AFM, allowed us to analyze the mechanical properties of matrices at the nanoscale level (curvature radius of the probe of 3.5 μm) where interactions between cells and fibers occur. This method is well suited for the study of soft biological specimens such as matrices or cells in liquid media. Previously published results report ECM stiffness values within the range of 2–150 KPa [24,38]. Our results demonstrate a relatively large scatter of values between the samples (Figure 4B). Thus, the lungs and spleen were among the softest (~2.9 KPa and ~3.6 Kpa, respectively), while the ovary matrix had the highest values of Young’s modulus (~6.5 KPa). Liver and kidney matrices have demonstrated intermediate results with Young’s modulus of ~5.6 KPa and ~6.1 KPa respectively. High heterogeneity at the nanoscale between the matrices was observed in previous studies as well [38,39] and might reflect the presence of heterogeneity in the structural characteristics of the decellularized matrices. The high difference between the results of macro- and nanoindentation was supposed to be due to different areas of force application (square millimeters for macroindentation and square micrometers for nanoindentation) as well as the contribution of porosity and matrix components of different stiffness, such as the connective tissue capsule, in the case of macroindentation.

### 3.5. Repopulation of the Decellularized Matrices

We have recellularized the obtained matrices with breast adenocarcinoma cells of the MDA-MB-231 and SKBR-3 lines. These cell lines were initially obtained from pleural effusions of breast cancer patients. The choice of these two lines of breast cancer is due to their difference in the degree of differentiation and invasive potential. The MDA-MB-231 line belongs to the triple-negative type of breast cancer, characterized by an extremely low degree of differentiation, mesenchymal phenotype, high invasive potential, and high proliferative activity. The SKBR-3 line, in turn, is a highly differentiated cancer, morphologically not much different from the squamous epithelium, with a low growth rate and low invasive potential [40,41,42]. The obtained decellularized matrices were subjected to repopulation by injection of cancer cells followed by cultivation for a week, after which histological preparations were made and analyzed. The recellularized matrices were analyzed in terms of the following parameters: degree of population, cell morphotype, proliferative activity, and nuclear–cytoplasmic ratio.

#### 3.5.1. Liver

During the recellularization of the liver matrix (Figure 5A), cells of both lines were detected in the liver parenchyma forming cluster structures. It is important to note that the cells of the MDA-MB-231 line have shown a partial change in the morphotype from pronounced mesenchymal to pseudoepithelial; the degree of the population was relatively low (Figure 5D) and proliferative activity was weak. Cells of the SKBR-3 line were found in a single number, morphologically epithelial. 

#### 3.5.2. Lungs

Recellularization of the lung matrix by MDA-MB-231 cells demonstrated a high degree of repopulation (Figure 5A,D). Cells were found throughout the entire space of the matrix, attaching themselves both to the fibers of the respiratory department and the remnants of the vascular bed, forming strands. The cell shape was fibroblast-like, and the morphotype was predominantly mesenchymal. A large number of actively proliferating cells that migrate towards the alveoli were detected. In contrast, the cells of the SKBR-3 line formed large, carcinoid-like clusters, the shape of the cells was irregular, closer to polygonal, and the cytoplasmic part was twice as large as the nuclear one. 

#### 3.5.3. Spleen

After repopulation of the spleen matrix by MDA-MB-231 cells, they were detected along the inner perimeter of the capsule with a single local migration into the core part of the organ (Figure 5A). Cells formed loose clusters, and proliferative activity was low. In some cases, only nuclei were detected, which indicated the death of the cells introduced into the matrix. In the case of SKBR-3 cells, multiple clusters were found scattered throughout the parenchyma, the cells have shown rare signs of proliferation, and the shape was irregular, closer to polygonal. 

#### 3.5.4. Kidney

The recellularization of the renal DCL matrices demonstrated a medium degree of repopulation for the MDA-MB-231 line (Figure 5B,D). The cells formed a dense layer under the surface of the capsule with long strands deep into the cortical substance. In the case of SKBR-3 cells, the formation of pseudoepithelial clusters located in the cortical substance could be noted, the number of cells was small, and proliferative activity was low.

#### 3.5.5. Ovary

After recellularization of the ovarian matrix, cells of the MDA-MB-231 line were disseminated throughout the entire thickness of the organ, morphologically moderately mesenchymal, and signs of proliferation were detected (Figure 5B,D). In the case of the SKBR-3 cells, we have mostly found only nuclei and remnants of apoptotic bodies, which indicates very low efficiency of the process for this line.

#### 3.5.6. Collagen

To compare cell growth in native matrices with that in an artificial collagen scaffold, the cells were embedded into the collagen hydrogel during its preparation. In a week of cultivation, the number of cells of the MDA-MB-231 line was relatively low, the cell sizes were reduced compared to cells in native matrices, and the cell shape was pronounced fibroblast-like (Figure 5C,D). Cells of the SKBR-3 line were found in a single quantity, the shape was changed from polygonal to rounded, and no cell clusters were found. Cells of both cell lines in the collagen hydrogel behave quite intact; there were no signs of death or signs of active cell division.

To determine the metabolic rate of cells in the matrices of different organs, we calculated the nuclear–cytoplasmic ratio (N/C ratio) for each cell line which was about 0.5 for MDA-MB-231 and 0.3 for SKBR-3 (Appendix A). There were no notable changes in the N/C ratio depending on the type of matrix for cells of the same line, however, a significant difference was shown between the lines MDA-MB-231 and SKBR-3 in the liver and lung matrices. The higher N/C ratio was revealed for cells of the rapidly growing MDA-MB-231 line, which is well explained by their higher proliferative activity and aggressiveness compared to SKBR-3 cells.

### 3.6. Correlations between Matrix Properties and Cancer Cell Growth

Based on the recellularization results, it became noticeable that cells behave differently in different matrices. This is related mostly to their growth rate but some changes were found for morphology. Comparison of cellular preferences allows us to assert their strong dependence on cell line (Figure 5D), and, assumably, on the degree of cancer cell differentiation. At the same time, it should be noted that the effect of the matrix origin on cell growth is observed not only between cell lines but also within each of the lines. To determine which characteristic of the matrix may affect cell growth, the data on different parameters were gathered together (Table 2). 

Analyzing the data, the collagen hydrogel differs crucially in its characteristics from the native matrices of any of the organs studied. It cannot be considered a relevant model and was not considered further.

In the case of the cells of the MDA-MB-231 line, the greatest growth was observed in the lung and ovarian matrices, which have the largest pore diameter and at the same time, the lowest values of total stiffness. In the case of SKBR-3 cells, a similar trend can be assumed, however, in the case of local stiffness. Based on the above observation, we hypothesized that cells of mesenchymal phenotype require low rigidity and high porosity of the 3D scaffold for optimal growth due to lower tension of the fibrils and a large amount of free space for migration. The hypothesis is indirectly supported by histological examination data demonstrating a high level of dissemination of MDA-MB-231 cells in tissues (Figure 5). On the contrary, for cells of a more epithelial phenotype, as for SKBR-3, stiffness of the substrate is important precisely at the site of the cell interaction with matrix fibers, since they are initially limited in movement, and cannot migrate in search of a more convenient niche.

The assumption was tested with a correlation analysis (Figure 6). As we expected, a significant negative correlation (Pearson correlation coefficient r = −0.97, *p* = 0.0066) was discovered between the total matrix stiffness, measured by macroindentation, and the degree of its repopulation by MDA-MB-231 cells (Figure 6A). Also, a strong positive tendency (r = −0.84, *p* = 0.07) was revealed between the pore area and MDA-MB-231 repopulation (Figure 6B). At the same time, the degree of repopulation by SKBR-3 cells does correlate (r = −0.94, *p* = 0.0178) with the local stiffness, measured by nanoindentation (Figure 6C). No relation was observed between repopulation by MDA-MB-231 and local stiffness (Appendix A), or between repopulation by SKBR-3 cells and total matrix stiffness or its porosity (Appendix A). 

Taken together, two main findings should be underlined. First, the correlation analysis testifies that in the case of mesenchymal-like cells, the matrix architectonics, porosity, and total stiffness of the organ are more important for the success of proliferation than the subtle ultrastructural matrix features, while in the case of the cells of more epithelial phenotype, the opposite priorities are observed. Second, the low stiffness of the native matrix is a promoting factor for cancer cell growth.

## 4. Discussion

### 4.1. Decellularization Protocol for Murine Organs 

The tumor models based on decellularized organs are becoming extensively used in cancer research. They are designed to fill the gap between the simplified low-relevant in vitro models and tumor-bearing animals. However, despite the wide use of mice as experimental animals, there are only a few protocols for the decellularization of murine organs and those are adapted to individual organs of certain tissue structure, for example, cardiopulmonary complex [26], lungs [43], or muscles [27]. Here we propose a simple and versatile protocol of the decellularization, applicable to a spectrum of murine organs of different tissue structure. Using histological and SEM analysis and quantifying the residuals of DNA, we demonstrate that our protocol ensures an accurate preservation of architectonics of ECM and complete elimination of the cellular component. We obtained decellularized matrices of murine organs, including liver, lung, spleen, kidney, and ovary to be used as a tool to analyze the tissue-specific behavior of breast cancer cells in terms of cell preferences in matrix stiffness and porosity. 

As any other model, DCL matrices have a number of advantages and disadvantages. One of the most appealing features of the DCL matrix is the preservation of native tissue-specific microarchitecture and biomechanics. A number of studies have shown that in the life of normal and malignant cells, their three-dimensional arrangement relative to each other is critically important, both in the dormant state and in cell movement [42,43,44]. It is also important that DCL matrices retain the chemical composition of the native matrix [45], which ensures cell–matrix and cell–cell interactions that are close to real tissue, as well as the deposition of a number of signaling molecules and factors in the matrix [46,47]. High biocompatibility of DCL matrices allows for full-scale three-dimensional cultivation and co-cultivation of cell populations and high relevance simulation of the tumor microenvironment [48]. Nevertheless, like any model, DCL matrices have a number of disadvantages. Firstly, the model is quite labor-intensive to obtain, which reduces its applicability in high-throughput studies. Secondly, the changes in the structure and biomechanics of the matrix due to the decellularization procedure must be taken into account. For example, it has been shown that DCL protocols may lead to a reduced concentration of glycosaminoglycans [49], or modified stiffness of the matrix [50]. Also, the complex composition of the matrix increases the number of unknown variables, which influences the results and complicates their interpretation. Nevertheless, despite the limitations, DCL matrices provide an uncontested tool that allows creating high-level tissue-specific in vitro tumor models [51]. In this regard, the DCL matrices are a potent model for metastasis studies, recapitulating the basic set of tissue characteristics including three-dimensionality, biological identity of matrix-forming polymers, intercellular and cell–matrix interactions, depot of biologically active signaling molecules, volumetric diffusion, biomechanics, etc., and also allows for co-cultivation of cell lines of different origin.

### 4.2. Biomechanical and Structural Properties of DCL Matrices of Murine Organs

One of the most important parameters of the DCL matrix as a platform for tumor growth model is its biomechanical properties, in particular its stiffness. Numerous studies on 2D substrates have shown the role of stiffness in the regulation of vital processes of behavior of the cells including proliferation, migration, and adaptability to environmental conditions [52,53,54,55,56,57]. However, little is known about the reaction of cells to matrix stiffness in 3D tissue-derived matrices [12,58,59,60]. There is a shortage of experimental data both regarding the mechanical properties of the matrices themselves and regarding the specifics of the cell behavior in them.

The most commonly used method to measure the stiffness of the tissue matrix is nanoindentation. This method is based on the precise determination of the interaction point of the indenter and the sample, which allows for determining the depth of penetration [61]. In this case, the stiffness of individual fibers (local stiffness) is measured at the level of cell interaction with the ECM. Our measurements of Young’s modulus of the DCL-matrices by nanoindentation demonstrate that it lies at the range of 2.9–6.5 kPa (Figure 4B). The results obtained are consistent with the work of Jorba et al. who reported analogous Young’s modulus for native (1.96 kPa) and decellularized (1.6 kPa) rat lung tissue [36]. 

The second approach, macroindentation, allowed us to evaluate the macroscale, or total, stiffness of the tissue. This parameter is integral and depends on the organization of fibrils and the degree of their packing, as well as on the overall three-dimensional organization of all elements of the matrix. We have evaluated total stiffness of several types of DCL matrices by macroindentation, based on the measurements of deformation by compression, using the “Biomomentum” device. The obtained data indicate that the absolute values of the Young’s modulus measured by this approach range from 17 to 40 kPa (Figure 4A). Published values of the Young’s modulus differ greatly and make it difficult to compare them with each other. Here we summarize data on the Young’s modulus of the same decellularized organs we used in this work but from different organisms, taking into account the method by which the values were obtained (Table 3). The variability of these absolute values arises from the implementation of different measuring techniques and parameters, such as executing tensile or compression tests, using different indentors and different rates of measurements. 

We should highlight that the apparent consistency in the stiffness of the DCL matrices obtained by us and previously reported values for native tissues serves as an additional sign of high preservation of the matrix structure during the proposed decellularization procedure [36,63]. We must note that the stiffness of the collagen gel differs by an order of magnitude from the stiffness of native matrices. In addition, in artificial gel, collagen fibril packaging is denser than in native matrices, whereas the pore area is smaller (Figure 4A). So, the biomechanical properties of the native tissue cannot be imitated by using collagen and, assumedly, other individual matrix proteins solely.

### 4.3. The Current Ideas about Stiffness-Dependent Behavior of Cells on 2D and 3D Substrates

The stiffness-dependent behavior of both normal and tumor cells has been extensively studied on 2D models of tumor growth [71,72,73,74]. The cell preferences were analyzed for a variety of 2D substrates of both natural and synthetic origin, including those based on collagen, fibronectin, polyacrylamide, polydimethylsiloxane, and their derivatives [71,75,76,77,78]. In some cases, materials unusual for biological purposes, such as glass or rubber, were also tested [72,79]. For the wide range of investigated stiffness, from tenths to several hundreds of kPa [76,80], an increase in stiffness led to higher growth rate and mesenchymal morphology of tumor cells with few reported exceptions [75,81,82,83]. 

The aforementioned situation, however, becomes largely ambiguous when moving to cell behavior on tissue-mimicking three-dimensional models. For example, when using a simple multicomponent gel made of collagen and polyacrylamide, it was shown that cells of more than a dozen of breast cancer cell lines demonstrate a higher growth rate while having a more pronounced mesenchymal morphology in soft gels (0.5–8 kPa), compared to 2D models [83,84]. A similar situation with change in cellular morphology compared to 2D models was shown for hepatocellular carcinoma and cholangiocarcinoma cell lines in models based on hydrogels obtained from DCL matrices [18,85]. 

Several previous works have reported the stiffness-dependent behavioral features of the MDA-MB-231 cell line studied in our work. When using a DCL tumor matrix, increasing stiffness did not affect MDA-MB-231 cell line growth speed and morphology but led to the development of drug resistance [17]. In contrast, in the case of cell growth in the bone matrix, high stiffness contributed to both an increase in growth rate and an increase in mesenchymal morphology of the MDA-MB-231 cell line, similar to the results obtained in 2D models [86]. It has also been shown that the cells of the MDA-MB-231 line grow better in a stiffer-aged breast matrix compared to a younger soft one [71]. Despite seeming contradiction, these results are consistent with ours: the highest values of the breast matrix stiffness reach about 2 kPa that is close to the stiffness of the lung matrix obtained in our study with the highest tendency to be repopulated; thus, both the results evidence the existence of an optimal stiffness range for tumor cells’ growth. This heterogeneity of results leads to the idea that additional factors influencing tumor cells in a 3D matrix become determinative; cell behavior cannot be extrapolated from simple 2D models and requires further investigation.

### 4.4. Behavior of Breast Cancer Cells in DCL Matrices of Murine Organs with Different Biomechanical Characteristics

We analyzed for the first time, on a spectrum of normal murine organs, the influence of 3D matrix biomechanics on cell behavior in DCL matrices. These models were created to simulate the metastatic process and to reveal the involvement of biomechanics in determining the premetastatic niches. Two human breast adenocarcinoma cell lines were chosen, namely, MDA-MB-231 and SKBR-3. Both cell lines are widely used in experimental oncology to create xenograft tumor models. MDA-MB-231 is a low-differentiated tumor cell line with mesenchymal phenotype, high migration potential [87,88], and high growth rate of xenograft tumors in mice [89,90]. On the contrary, SKBR-3 is a highly differentiated tumor cell line with an epithelial phenotype [39], which demonstrates a low growth rate and invasiveness [91,92] and local growth without migration in xenograft models [93,94]. The peculiarities in cell phenotype and behavior were preserved in repopulated DCL matrices in our study. Thus, the MDA-MB-231 cells showed a high growth rate and invasiveness in all of the obtained matrices; the cells retained their mesenchymal phenotype and were reluctant to clusterization (Figure 5). SKBR-3 cells showed a low growth rate and invasiveness, high clusterization, and retained their epithelial phenotype (Figure 5). 

Quantitation of the growth of the studied cell lines in the matrices of various organs revealed that MDA-MB-231 cells actively populated all matrices of the lungs, liver, kidneys, spleen, and ovary within a week of observation but with a two-fold difference in resulting total cell amounts between the least and most populated matrices. SKBR-3 cells showed a low growth rate in lung, kidney, and spleen matrices, with further decrease in the liver matrix and a lack of growth in the ovarian matrix (Table 2). 

According to medical statistics, metastasis of certain types of cancer occurs predominantly in fairly specific organs. For example, the targeted organs and tissues for breast cancer are the lungs, liver, and bone tissue [95]. However, in each individual case, the location of the metastases can vary depending on the genetic/metabolic profile of the individual tumor [95,96,97]. A number of studies demonstrate that breast cancer cells are capable of metastasis to multiple non-specific tissues, for example, to gynecological organs [98,99,100]. In addition, depending on the origin of the cells (primary tumor node or metastatic focus), the targeted tissue may vary.

Putting together our findings regarding tumor cells’ growth preferences with the data on the stiffness of the matrices, one can see peculiar patterns confirmed by correlation analysis. MDA-MB-231 cells, which are of mesenchymal phenotype, proliferate most successfully in matrices with low total stiffness and large pore size (Figure 6A,B). SKBR-3 cells, which are of epithelial phenotype, grow more successfully in matrices with low local stiffness, regardless of pore size (Figure 6C). Our data on the behavior of MDA-MB-231 cell lines are consistent with previously reported evidence that the stiffer matrix slows down the proliferation of MDA-MB-231 cells while increasing their stemness [101]. Thus, it can be assumed that in the case of invasive cells lines, the large pore size of the matrix is crucial as it does not interfere with cell migration [102,103], and for non-invasive lines, local stiffness is an important factor, since growth in the colony involves the spreading of matrix fibers during cell growth [104,105]. 

The early observation that organ distribution of metastases of breast cancer was non-random gave rise to a famous “seed and soil” theory of metastasis [106]. To date, specific metastatic sites have been established for various types of breast tumors [107]. The “seed and soil” theory is now reinforced with the concept of the pre-metastatic niche (PMN). PMN is a predetermined microenvironment in a distant organ that was preconditioned by the primary tumor to provide the survival of arriving metastatic cells [108]. PMN formation is determined by a number of different factors. The leading role in PMN initiation is played by tumor-derived signaling molecules [109] and extracellular vesicles [110,111], which maintain a complex communication between normal and malignant cells. The high leakiness of vasculature, contributing to hypoxic status in the target site, and some other tissue-specific conditions are also of high importance [112]. The named above factors are also responsible for recruiting immune cells to the PMN [113], which leads to the establishment of an inflammatory [114,115] and immunosuppressing [116] microenvironment. The final stage of PMN formation is connected with remodeling the ECM [117], which includes changing its composition and stiffness.

The main components of ECM which determine its stiffness are various collagens and hyaluronane [118]. Thereby, the most crucial cellular receptors for sensing matrix stiffness are integrins and CD44 (hyaluronic acid receptor). Integrins implement their function as a part of adhesion complexes containing multiple adapter proteins including kinases and GTPases [119]. Under force application, these complexes undergo dynamic changes which initiate the signal transduction. Integrin-mediated signals can regulate cell behavior by controlling the cell cycle by the FAK/Rac [120], PI3K/Akt, Mek/Erk [121], and Hippo [122] signaling pathways. The regulation of cell migration mediated by integrins is realized by the upregulation of various metalloproteinases, hindering adherens junctions formation, and providing amoeboid single-cell migration [123]. Hyaluronic acid receptor CD44, in turn, can regulate cell proliferation and invasion by Wnt/β [124], Hippo [125], and RAS/FAK/ERK [126] signaling pathways.

The patterns revealed in this work demonstrate the inverse dependence of the breast cancer cells on 3D matrix stiffness that is opposite to cell preferences on 2D substrates. We have proved this dependence for the first time on DCL matrices of normal murine organs and hypothesize that the ECM architecture and biomechanics are important players in guiding the metastatic process along with chemical factors. 

It is quite obvious that even in cases of elimination or minimizing the impact of the majority of the factors, which influence cells in a living organism, the behavior of cells in a relatively pure DCL matrix is still determined by the cumulative interaction of various parameters, both biomechanical and others. We must note that in fact, the influence of individual factors can be both equally or oppositely directed (for example, mechanotaxis may oppose chemotaxis) [14,127,128]. In order to elucidate the role of each of these factors, in the present work, we made an attempt to isolate one specific factor and analyze its impact on breast cancer cell proliferation and repopulation of the matrix. This does not negate the fact that in the complex conditions of a full-fledged TME, the summation of the factors in the microenvironment context will determine the resultant cellular response.

We recognize that the expansion of the list of tested organs in the following studies is undoubtedly required to test the hypothesis. Another factor to be taken into account is phenotype-specific cell behavior and peculiarities arising from cell origin, metabolic activity, and cell–cell interaction. Co-culturing of tumor cells with cells characteristic of tumor microenvironment, including endothelial cells, fibroblasts, and a subset of immune cells could meet some of the issues [129] and further elucidate the cells’ behavior in a complex tumor environment.

## 5. Conclusions

In this work, we have proposed the new decellularization protocol and proven its universality and effectiveness for a spectrum of murine organs. The biomechanical properties of DCL organs were found to play an important role in determining the fate of cancer cells and their progression. We showed for the first time the influence of the biomechanical features of a three-dimensional model based on the DCL matrix of various murine organs on the behavior of breast cancer cells. We found strong correlations between the growth of cells of different phenotypes and matrix parameters. Thus, cells of mesenchymal phenotype (MDA-MB-231) prefer a matrix with lower total stiffness and larger pores, while cells of the epithelial phenotype (SKBR-3) need softer local stiffness at the fiber level. Hence, the model we develop can illuminate the processes underlying tumor development, invasion, and metastasis, particularly the potential contribution of ECM to the development of pre-metastatic niches.

## Figures and Tables

**Figure 1 cells-12-02030-f001:**
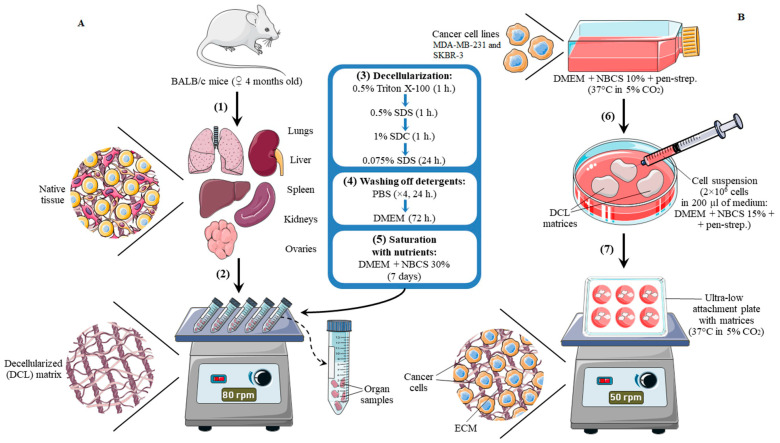
Simplified scheme of decellularization (**A**) and recellularization (**B**) procedures. The liver, kidneys, ovaries, spleen, and lungs were collected (1) from mice of BALB/c line in sterile conditions. The collected organs were washed in sterile distilled water. Fat adhesions were removed from each organ, after which the organs were dissected into 0.3 × 0.3 cm pieces and placed in 50 mL tubes (2). For decellularization (3), the samples were subsequently incubated in Triton X-100, sodium dodecyl sulfate (SDS) solution, sodium deoxycholate (SDC) solution, and again in SDS solution. At each stage of the protocol, the tubes with samples were fixed on an orbital shaker with a rotation speed of 80 rpm. After decellularization, the samples were washed off the detergents (4) with PBS and DMEM media and saturated with nutrients (5) in DMEM media with 30% newborn calf serum (NBCS) for 7 days with a change of medium every 72 h. For recellularization, each preconditioned matrix was placed in an individual well of a 6-well ultra-low attachment plate and repopulated by injecting 300,000 cells in 500 μL of complete growth media (6). The cell suspension was distributed inside the matrix by several injections via an insulin syringe; 5 mL of DMEM with 15% serum was added to each well of the plate, and the matrices were incubated at 37 °C in an atmosphere of 5% CO_2_ for 7 days (7). On days 3 and 5, the medium bathing the matrix was collected and the cell pellet was re-injected into matrices.

**Figure 2 cells-12-02030-f002:**
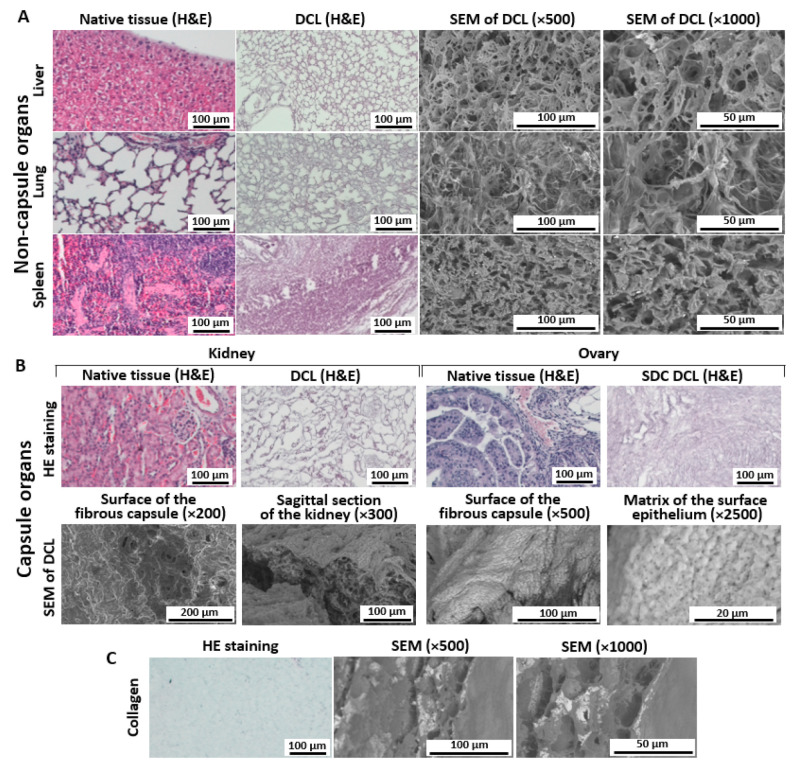
Histomorphological and SEM analysis of decellularized matrices obtained from non-capsule organs (**A**), capsule organs (**B**), and a collagen matrix (**C**). Representative images from 20 to 50 samples are shown. For histological analysis, hematoxylin–eosin staining was used. Two different magnifications for SEM images are provided in (**A**,**C**) to visualize the degree of integrity of the matrix and the morphology of fibers. For capsule organs (**B**), SEM images were obtained for different parts of the organ to represent variability in structure. See explanations in the text.

**Figure 3 cells-12-02030-f003:**
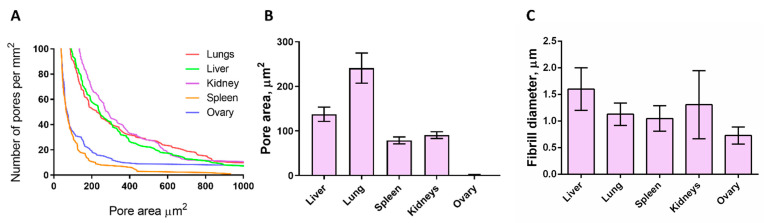
Microstructural characteristics of the decellularized matrices. (**A**) Pore area distribution for matrices of different organs obtained by histological image analysis (*n* = 20–50). (**B**) Averaged pore area for matrices of different organs calculated from SEM images (*n* = 20–50). For details of statistical comparison between matrices, see Appendix A. (**C**) Fibril diameter for matrices of different organs measured by SEM (*n* = 20–50). For details of statistical comparison between matrices, see Appendix A.

**Figure 4 cells-12-02030-f004:**
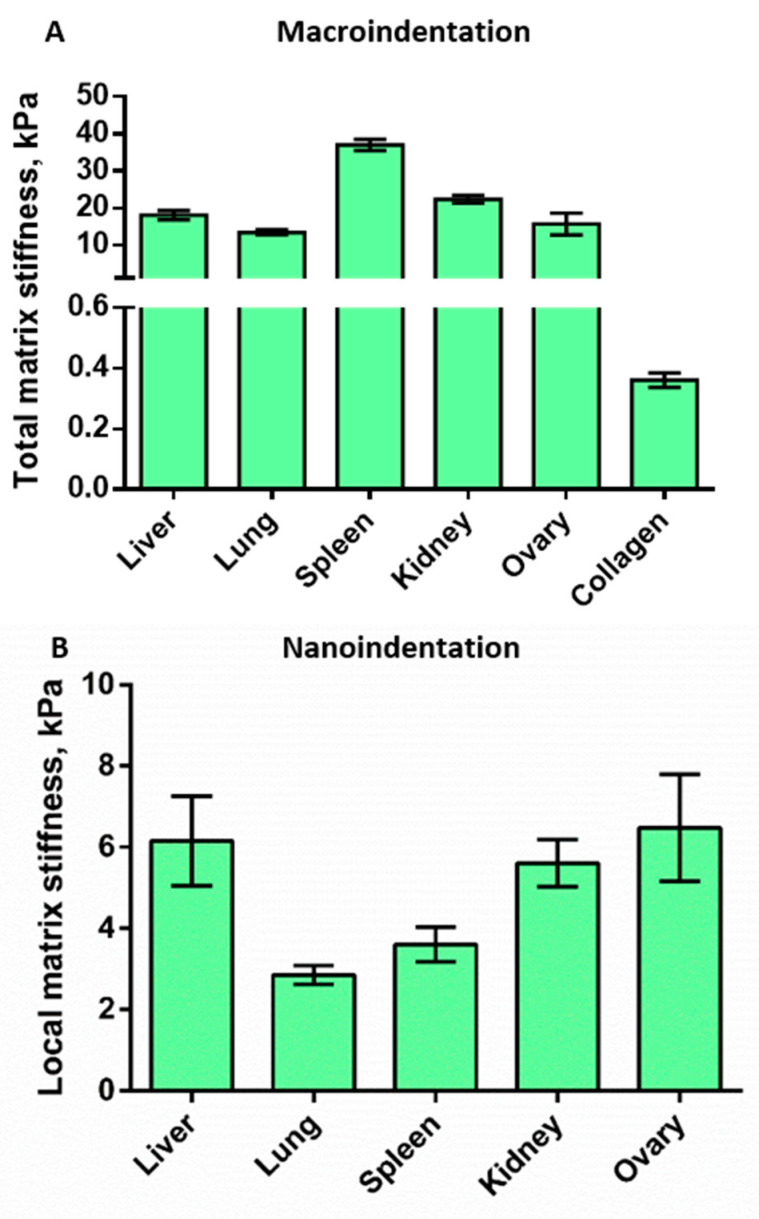
Stiffness characteristics of matrices measured by macroindentation (**A**), and nanoindentation (**B**). (**A**) Total matrix stiffness, measured by macroindentation. A decrease in Young’s modulus in the series spleen > kidney > liver > ovary > lung ≫ collagen was observed (*n* = 20). For details of statistical comparison between matrices, see Appendix A. (**B**) Local matrix stiffness measured by nanoindentation. A relatively large difference between all the samples was demonstrated with the lung being the softest (*n* = 25–30). For details of statistical comparison between matrices, see Appendix A.

**Figure 5 cells-12-02030-f005:**
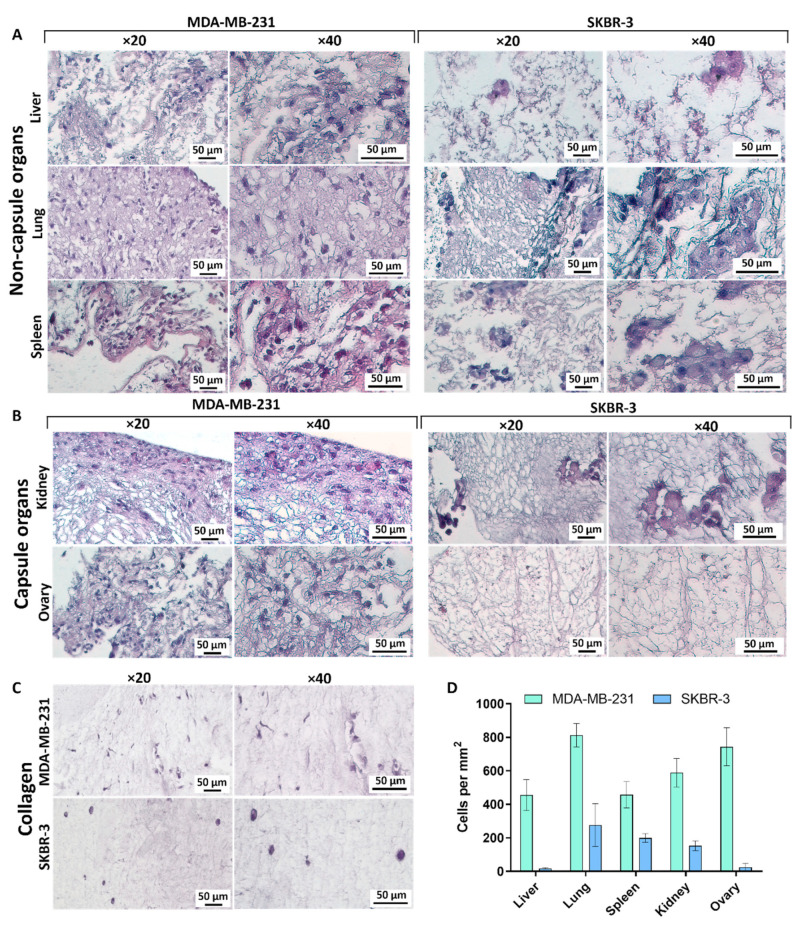
Histomorphological analysis of recellularized matrices of non-capsule organs (**A**), capsule organs (**B**), and a collagen matrix (**C**). Representative images from 20 to 50 samples are shown. For histological analysis, hematoxylin–eosin staining was used. Two different magnifications are provided to visualize the degree of the overall population of the matrix and the morphotype of individual cells. See detailed explanations in the text. The quantitative assessment of the repopulation of the matrices is given in (**D**).

**Figure 6 cells-12-02030-f006:**
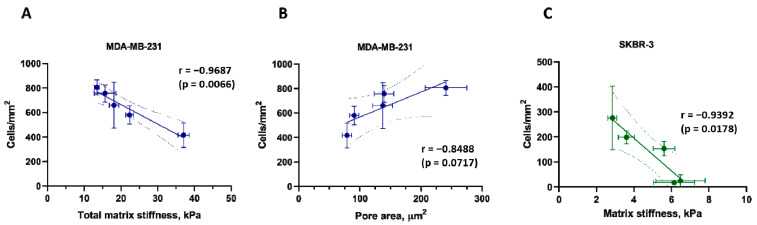
Analysis of the correlation between the matrix properties and the growth of the repopulating breast adenocarcinoma cells (*n* = 20–30). (**A**) Total matrix stiffness vs. the number of MDA-MB-231 cells. (**B**) Pore area vs. the number of MDA-MB-231 cells. (**C**) Local matrix stiffness vs. the number of SKBR-3 cells. The dotted lines indicate the 95% confidence interval for the linear regression trend. The Pearson correlation coefficient (r) with a significance level (*p*) are indicated for all the comparisons.

**Table 1 cells-12-02030-t001:** DNA content in native and decellularized tissues of murine organs.

	Total DNA Content		
	Native Tissue, µg/g Wet Tissue Weight	DCL Tissue, µg/g Wet Tissue Weight	DCL Tissue, % from Native
**Liver**	1032 ± 11	0.63 ± 0.28	0.06
**Lung**	1955 ± 29	0.90 ± 0.12	0.05
**Spleen**	2337 ± 30	2.12 ± 0.14	0.09
**Kidney**	1286 ± 45	1.03 ± 0.32	0.08
**Ovary**	1757 ± 63	1.22 ± 0.26	0.07

**Table 2 cells-12-02030-t002:** The main characteristics of the decellularized tissues and cell populations in different matrices.

		Liver	Lung	Spleen	Kidney	Ovary	Collagen Gel
Matrix fiber density		Medium	Low	High	Medium	High	High
Pore area, μm^2^		133	240	78	90	152,34	N/A
Local matrix stiffness (nanoindentation), kPa		6.15	2.85	3.6	5.6	6.4	0.36
Total matrix stiffness (macroindentation), kPa		18	13.4	37	25.4	15.7	N/A
Degree of repopulation	MDA-MB-231	Medium	High	Medium	Medium	High	Low
	SKBR-3	Single	Low	Low	Low	Single	Low
Cell morphology	MDA-MB-231	Mes ^a^/Ep ^b^	Mes	Mes	Mes	Mes	Mes
	SKBR-3	Ep	Ep	Ep	Ep	Ep	Ep

^a^—mesenchymal morphotype; ^b^—epithelial morphotype. The color coding from red to yellow and green is used to highlight the increase of a parameter. Gray is used to highlight the headings, blue to indicate the mesenchymal cell morphology, purple for the epithelial cell morphology.

**Table 3 cells-12-02030-t003:** Values of Young’s modulus of decellularized organs.

DCL Organ	Young’s Modulus	Method	Device and Measuring Conditions	Ref.
Porcine liver	1.25 ± 0.07 kPa	Compression test	Zwick/Roell ProLine Z005 testing machine (Zwick/Roell, Ulm, Germany) equipped with a 10 N load cell.	[62]
Ferret liver	1.18 kPa	Compression test	ElectroForce TestBench mechanical testing system equipped with a 1000-g load cell (Bose ElectroForce, Eden Prairie, MN, USA).	[63]
Rat liver	145 ± 19.68 kPa	Tensile test	Uniaxial tensile testing machine (CMT8502, Shenzhen New Sans Test Technical Company, Nanshan, Shenzhen, China). The test was performed at loading speed of 1 mm/min until the final fracture of the specimen.	[64]
Human liver	18,490 ± 1400 kPa	Tensile test	Instron 3367 dual column universal testing system (Instron, Glenview, IL, USA) fitted with Instron biopulse submersible pneumatic side action grips (Instron, USA) and a 50 N load cell. A gauge length of 20 mm and an extension rate of 20 mm/min were used.	[65]
Rat lung	~80 kPa	Tensile test	Santam tensile testing machine STM-1 (SANTAM, Tehran, Iran). The uniaxial tensile test was carried out by a crosshead speed of 2 mm/min, elongation raised to 20–40 mm, and a load cell of 6 kg at a constant elongation speed until rupture of the specimen was observed.	[66]
Rat lung	74.91 ± 5.78 kPa	Tensile test	Material testing machine BZ2.5/TN1S (Zwick/Roell, Ulm, Germany). Preloading of 0.015 N was imposed and the sample length was reported, then a preload of 0.003 N was set.	[67]
Rat lung	0.38 ± 0.07 kPa	Tensile test	Displacement actuator 300C-LR (Aurora Scientific, Aurora, ON, Canada) with a force transducer 404A (Aurora Scientific, Canada). Uniaxial tensile measurements after preconditioning with triangular stretch at 0.1 mN force.	[68]
Porcine kidney (cortex)	6.4 ± 2.7 kPa	Compression test	Instron 3342 Single Column Universal Testing System (Instron, USA) and Instron Model 1321 (Instron, Glenview, IL, USA) were used to perform compression tests at a rate of 0.07 mm/sec until a compression force of 45 N was reached.	[69]
Rat Kidney	178.9 ± 50.2 kPa	Tensile test	Tensile mechanical testing machine (Instron, Glenview, IL, USA). Samples were preconditioned by cycles of loading and unloading. Strain rate was 0.01 s^−1^ throughout the test.	[70]
Rat liver	4242.2 ± 891.6 kPa			
Rat Lung	233.6 ± 98.0 kPa			

## Data Availability

The data presented in this study are available on request from the corresponding authors.

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
