# Peer review of "Breast Cancer Cell Type and Biomechanical Properties of Decellularized Mouse Organs Drives Tumor Cell Colonization"

_cells, 2023, doi:10.3390/cells12162030_

Round 1

Reviewer 1 Report

The authors prepared decellularized ECM from various tissues that are main metastatic targets of breast cancers and their rare targets. These decellularized ECM possess different mechanical properties. The authors showed that breast cancer cell lines exhibited different growth rate on the decellularized ECM according to their mechanical properties. Overall, the authors well-performed the experiments. However, I felt that some conclusions are not fully supported by the data. Also, the discussion from the viewpoint of cell biology is lacked. So, I recommend not publish this manuscript in your journal and encourage to transfer to your sister journal “Materials”. Specific comments are below.

(Major comments)

1)     As the authors described, liver and lung were selected as the main metastatic targets of breast cancers and spleen, kidney, and ovary were selected as the rare targets. However, MDA-MB-231 cells were highly grown on decellularized lung, kidney, and ovary although kidney and ovary are not main targets of breast cancer metastasis. Why? Is it suggested that the metastasis targets are not decided by mechanical properties? The authors should discuss the reasons.

2)     Moreover, the authors should conclude the feasibility of their decellularized ECM as ECM models for studying cancer metastasis, based on the discussion in submitted manuscript and above questions. Also, I believe that the description of advantages and disadvantages will increase the value of this manuscript.

3)     The authors claimed that the cell growth was highly correlated with decellularized ECM stiffness. However, the authors just compared the growth on decellularized ECM prepared from different tissues with different mechanical properties. The ECM from different tissues possess different compositions, leading to different intracellular signaling and cell behaviors. To exclude the effects of ECM compositions, the authors should check the growth on crosslinked decellularized ECM that should show stiffer properties.

4)     I was just wondering whether decellularized ECM prepared from breast or adipose tissues were necessary as an experimental control. Because these tissues are the original tissues of the cells used in this study.

(Minor comments)

1)     Please check subsection number and title of 2.6.

2)     Please check the units in section 3.2.5 (line 347-352). Micrometer is correct?

3)     Please show the method to prepare collagen matrix. And please show SEM image of collagen matrix with higher resolution.

4)     The order of X-axis labels is correct in Figure 3C? Why did the authors order in “liver”, “lung”, “kidneys”, “spleen”, and “ovary” in only Figure 3C.

5)     Please describe the method how to count cell numbers in Figure 5D.

6)     Please use subsections in Discussion part. Present discussion part is too long.

Author Response

Dear Reviewer,
We would like to express our sincere appreciation for your careful attention to our manuscript and for the suggested improvements and valuable comments. We have revised the manuscript according to your remarks. Please find  the detailed description of the revision with our point-by-point answers in the attached file.

Reviewer 2 Report

The authors nicely present their work on colonisation of two breast cancer cell lines with different properties, in different decellularized mouse organs. The manuscript is well written and easy to read. This work gives insightful information about biomechanical properties of different organ scaffolds and how breast cancer cells respond to this.

Overall I think this is good work, but as expected from me, I also have some (minor) comments to improve the manuscript.

- I do not particularly like the title. It doe not seem right. Maybe consider to rephrase the title and start with for instance: Breast cancer cell type and biomechanical properties of decellularized mouse organs drives tumor cell colonisation.

- In the abstract you introduce the abbreviation DCL for decellularized tissues. What is the L in this abbreviation? I get the D (decellularized)... Is it really necessary to make an abbreviation for this? the more abbreviations, the less easy reading.

- Overall English grammar is good. However, here and there I found some mistakes. One is 'mice' on page 3, line 119 - which in my sight should be 'mouse'. Page 4, line 161 heading of 2.6.3. What is the 'D'?, page 6, line 270: the first sentence seems not to be right. And a few more - so my advice is to critically read through the whole manuscript for small grammar flaws.

- a major issue is that I do not find any information about replicates. How many mice were used? How many organs were procured and analysed? There is a P value here and there, but no sign of numbers of replicates done. Please add this to the materials and methods section and in the figure/table legends (where appropriate).

- In table 1, English numbering should be shown. Instead of 1757,1 it should be 1757.1. Related to this, why is the one decimal only shown in case of the ovary and not for the other organs? And what is the ug/g mean?

- possibly my printer quality is poor, but I had some difficulty really seeing detail in Figure 2.  and 5. Maybe good to add a zoom-in detail. And, related to the replicates, what about it in this analysis? Are these representative for all other tissues?

- The analysis shown in Figure 3. might be good to show in moe detail in the Supplemental Information. How did you analyse this in ImageJ? What was considered a pore for instance? Good to show the strategy here.

See above - overall good, but take a bit of time to go through the whole text specifically with English grammar and style in mind.

Author Response

(The authors gave the same response as above.)

Round 2

Reviewer 1 Report

The authors answered the questions properly.

Now, I can recommend to publish the manuscript as it is.